# Task Assignment Optimization in Multi-UAV-Assisted WSNs Considering Energy Budget and Sensor Distribution Characteristics

**DOI:** 10.3390/s23187842

**Published:** 2023-09-12

**Authors:** Qile Xie, Wendong Zhao, Cuntao Liu, Laixian Peng

**Affiliations:** 1College of Communication Engineering, Army Engineering University of PLA, Nanjing 210007, China; xql2022wt@163.com; 2Military Subarea of Da Hinggan Ling, Da Hinggan Ling Prefecture, Da Hinggan Ling 232700, China; liucuntao@aliyun.com

**Keywords:** wireless sensor networks, mission completion time, unmanned aerial vehicles, data collection

## Abstract

In emergency situations, such as disaster area monitoring, deadlines for data collection are strict. The task time minimization problem concerning multi-UAV-assisted data collection in wireless sensor networks (WSNs), with different distribution characteristics, such as the geographical or importance of the information of the sensors, is studied. Our goal is to minimize the mission time for UAVs by optimizing their assignment, trajectory, and deployment locations, while the UAV energy constraint is taken into account. For the coupling relationship between the task assignment, trajectory, and hover position, it is not easy to solve the mixed integer non-convex problem directly. The problem is divided into two sub-problems: (1) UAV task assignment problem and (2) trajectory and hover position optimization problem. To solve this problem, an assignment algorithm, based on sensor distribution characteristics (AASDC), is proposed. The simulation results show that the collection time of our scheme is shorter than that of existing comparison schemes when using the same data size.

## 1. Introduction

Currently, small unmanned aerial vehicles (UAVs) have shown promising application prospects in assisting wireless sensor networks (WSNs) with data collection and processing due to their high maneuverability, low cost, and no risk of human casualties [1,2,3,4]. Through the line-of-sight link between the UAV and the sensor devices or UAVs, high-speed data transmission can be achieved, which helps to meet the requirements of sensor data collection and transmission [5,6]. Moreover, UAVs have good environmental adaptability and operational flexibility. When the sensor devices are located in areas without network coverage, UAVs can be quickly deployed, on-demand, to expand the coverage range of the available network, thereby addressing the problem of communication failure between the sensor devices and the data center, and achieving efficient data collection regarding the sensor devices [7,8,9].

Due to factors such as onboard battery capacity, payload capacity, and communication capabilities, the data collection capacity of a single UAV is limited. In scenarios where there is a large number of sensors, distributed over a wide area, or if the devices are situated far away from the network coverage area, a single UAV typically encounters challenges in fulfilling the necessary data collection time requirements. In response to this, the application of UAVs in auxiliary data collection is gradually moving towards multi-UAV cooperation [10,11,12,13,14,15].

In some emergency situations, such as disaster area monitoring, geological surveys, and industrial emergencies, there are strict deadlines for completing data collection tasks, which poses a challenge to the rapid and efficient flight and data collection of UAVs. In order to shorten the task completion time, several related studies have been carried out [13,15,16,17,18,19,20,21,22,23,24,25,26,27,28]. For clarity, the comparison of the related works and our work is presented in Table 1. For larger collection areas, the application of multi-UAV collaborative collection technology can shorten the task completion time. In this paper, the task completion time of the multi-UAV refers to the maximum task completion time among multiple UAVs. To achieve the goal of shortening the length of time taken to complete the task, all tasks need to be allocated as evenly as possible among the UAVs.

In the aforementioned study, the optimization of task assignments, the hovering point, and trajectory position has been widely discussed. When studying the generalizability of research problems, many authors often make the assumption that sensors are randomly distributed. However, in practical surveillance scenarios, UAVs exhibit specific distribution characteristics based on the nature of the monitoring targets. For instance, in forest environments, sensors are deployed based on the distribution patterns of vegetation. When the sensors exhibit partially dense and partially sparse characteristics within the collection area, UAVs need to be assigned in accordance with the geographical distribution characteristics of the sensor. In the case of densely distributed sensors, the number of sensors visited by the UAVs will increase, thereby increasing the collection time of the UAVs. Conversely, for sparsely distributed sensors, the distance covered by the UAVs to visit the sensors increases, resulting in an increased flight time for the UAVs.

Moreover, the amount of data provided by the sensors, and the distance between the UAVs and the sensors, will also affect the UAVs’ collection time, which comprises part of the task time. Therefore, we studied the collaborative task allocation of multiple UAVs for data collection, and we proposed a UAV task allocation scheme based on sensor location features. This scheme enables the data center to assign tasks based on sensor distribution and data size, thereby achieving the goal of minimizing task completion time. Through a performance comparison of the Genetic algorithm (GA) for solving the multi-traveling salesman problem (MTSP), we have demonstrated its effectiveness.

In this paper, we focused on the collaborative task allocation for data collection in a multi-UAV-assisted sensor network, considering the distribution characteristics of the sensors, in order to jointly optimize task association, collection trajectories, and the hovering points of UAVs to minimize the task completion time. The task completion time is decomposed into two sub-problems, as follows: (1) the UAV task assignment based on geographic information, and (2) the optimization of trajectories and hovering points. The main contributions are summarized, as follows.

For multi-UAV-assisted sensor network data collection, the problem of minimizing the task completion time is formulated as an optimization problem and decomposed into three sub-problems, as follows: (1) task assignment of UAVs based on the sensors’ geographic information and characteristics, and (2) the optimization of the trajectories and hovering points of each UAV.A characteristic-based clustering algorithm was used to classify the sensors, and the classified sensors were then assigned to the UAVs. For the collection of sensors in dense areas, a hybrid algorithm was used to optimize the hovering and collection positions of each UAV. For the optimization of the collection trajectory, the problem was transformed into the traveling salesman problem (TSP), and an ant colony algorithm was used to solve it.The simulation results of three sensors with different geographical distribution characteristics show that the proposed algorithm is superior to the comparison algorithm regarding collection time and energy consumption.

The rest of this paper is organized as follows. In Section 2, we introduce related works. The system model and problem formulation are presented in Section 3. Then, we propose the AASDC to solve the task assignment and data collection trajectory of UAVs in Section 4. Simulation results are provided and analyzed in Section 5. The discussion is provided in Section 6. Finally, we conclude the paper in Section 7.

## 2. Related Work

The high mobility of UAVs makes data collection freer and more efficient. In order to shorten task completion time, the flight path of UAVs can be optimized to obtain the shortest flight distance. Li et al. proved that as long as the UAV is within the transmission range of the sensor, it can successfully collect data [16]. Kim et al. regarded the collection area as a two-dimensional plane, and they utilized geometric mathematical methods to reduce the UAV flight distance [17]. Deng et al. proposed a new approximate algorithm that transformed the problem concerning the minimum trajectory length of moving points into a problem concerning minimum–maximum circle coverage with a domain, in order to minimize the longest travel [18]. By jointly optimizing the UAV’s trajectory and sensor allocation to minimize the UAV’s task completion time, Yuan, et al. put forward the optimal solution concerning UAV trajectory optimization, using the flight-hover mode in data collection [19].

However, designing the shortest flight path could lead UAVs over the transmission area boundary, which can result in increased air-to-ground path loss, and a reduced transmission rate. Therefore, optimizing the hover position is essential to reducing collection time. Lou et al. identified the maximum independent region set, and it was disjointed and maximized to obtain the minimum collection time [20]. Wang et al. proposed hover and flight modes for data collection, and they developed a UAV-sensor association mechanism and collection sequence mechanism to minimize the collection time [21]. Li et al. suggested using a V-shaped trajectory to optimize the collection trajectory of UAVs in hover-flight mode, thus minimizing the completion time of UAV missions [22]. Xu et al. studied the scenario concerning the deployment of the minimum number of UAVs to complete the data collection task under the task time constraints. The problem of UAVs flying directly above sensors, or within the transmission range for data collection, is considered [23].

To address the task allocation problem in a multi-UAV system, Qin et al. proposed a modeling approach using an undirected graph. The model assigns weights to the edges which represent the flight time, and the hovering points which indicate the collection time. Then, they transformed this problem into the minimum–maximum circle coverage problem, which they solved utilizing the K-circle algorithm [24]. Additionally, Zhan and Zeng offered two data collection modes, and they derived data collection points. Using the min–max MTSP, they assigned UAV tasks. Then, the UAVs’ flight path, hovering position, and wake scheduling of ground sensors were jointly optimized to minimize the maximum task completion time among all UAVs [25]. Zhang, et al. studied the problem of the minimum total flight time for UAVs in a rechargeable scenario. The sensor nodes were divided into multiple clusters using Gaussian mixture model clustering, which reduces the complexity of the task [15].

In addition, regarding the problem of UAV data collection, there are some factors worth considering. Shen et al. considered the UAV’s energy budget and cache capacity, as well as the data transmission constraints of IoT services, when optimizing the UAV’s trajectory and hover position to minimize the UAV’s data and total running time [26]. Liu et al. considered the different levels of importance and quality of service requirements of sensors in data collection [27]. Luo et al. proposed an efficient iterative algorithm to optimize 3D path planning and time allocation for UAV [13]. Hsu and Gau focused on the collision avoidance algorithm, based on RL, to obtain an optimal trajectory of UAV [28].

## 3. System Model and Problem Formulation

### 3.1. System Model

As depicted in Figure 1, our study focuses on the scenario concerning sensor data collection, with the assistance of a multi-fixed-wing UAV. In this scenario, ground sensors (GSs) are strategically distributed throughout the forest to acquire status information about the surrounding environment. Regarding the distribution of GSs, we considered the geographical location characteristics of sensors. The data center sends UAVs to collect data from the ground sensors, and it sends them back to the data center for analysis. For convenience, the major notations used in this paper are summarized in Table 2.

The collection of sensors and the collection of UAVs are denoted as N={1,…,N} and M={1,…,M}, respectively. In each data collection cycle, the UAV m∈M successively visits the sensors, $N_{m}$, in a subset of N, as assigned by the data center. Where $N_{m}=\{1,\ldots ,N_{m}\}$, the ith sensor of $N_{m}$ is denoted as $n_{m}^{i}$, and it represents the ith sensor visited by m within a task cycle. Its serial number in N is denoted as ${\tilde{n}}_{m}^{i}$.

The coordinate of the base station is $l_{BS}=[x_{BS},y_{BS},0]$, the coordinate of the sensor, n, is $l_{n}=[x_{n},y_{n},0]$, the instantaneous coordinate of the UAV, m, is $l_{m}(t)=[x_{m}(t),y_{m}(t),H]$, where H is the flight height of the UAV.

### 3.2. Channel Model

The channel model between the UAV and sensor was modeled as a line-of-sight (LoS) channel. In a data collection cycle, the moment when m starts to collect $n_{m}^{i}$, and the moment when it ends to collect sensors, is $t_{m}^{i,0}$ and $t_{m}^{i,1}$. The power when n∈N sends data to m is $P_{n}$, and the data transmission rate with the UAV remains constant in the process of data transmission. Then, the data transmission rate can be expressed as follows:(1)$$R_{{\tilde{n}}_{m}^{i}}(t)=B\log_{2}(1+P_{{\tilde{n}}_{m}^{i}}h_{{\tilde{n}}_{m}^{i},m}(t)/\sigma_{0}{}^{2}),$$
where $h_{{\tilde{n}}_{m}^{i},n}=\beta_{0}/{\left(d_{{\tilde{n}}_{m}^{i},m}(t)\right)}^{2}$ is the instantaneous channel gains between the UAV, m, and sensor, $n_{m}^{i}$. B is the available channel bandwidth, $\beta_{0}$ denotes the average channel power gain at the reference distance, $d_{0}=1m$. $d_{{\tilde{n}}_{m}^{i},m}(t)$ is the instantaneous distance between UAV, m, and sensor, $n_{m}^{i}$, and $\sigma_{0}{}^{2}$ denotes the white Gaussian noise power at the receiver.

### 3.3. Energy Consumption Model

To guarantee that the UAV can complete the mission with energy constraints, we use the following equation:(2)$$\int_{0}^{t_{n}+t_{c}}P_{f}(v)dt+t_{c}P_{c}\leq E_{\mathrm{th}},$$
where $E_{th}$ is the energy threshold of the UAV, $P_{f}(v)$ is the fight power of the UAV at speed, v (see [29], Equation (12) for details), $P_{c}$ is the circuit power consumed by UAV during data collection, and $t_{n}$ and $t_{c}$ are the total flight time and total data collection time of the UAV, respectively. Therefore, the data collection period of the UAV, m, can be expressed as follows:(3)$$T_{m}=t_{f_{m}}+t_{c_{m}},$$

### 3.4. Data Collection Model

The UAV was equipped with K directional antennas, and it was assumed that the sensor distance during the experiment was far enough so that the directional antennas would not interfere with each other. The set of the channel collection is expressed as K={1,…,K}. The UAV can simultaneously access K sensors in the transmission range. The location information of the static sensor can be get before the UAVs take off from base station. The sensors accept the activation signal from the UAV directional antenna, and they send the data stored in the sensor through the all-directional line. The UAVs receive the sensors’ data via a directional antenna, and the interference is ignored in the state of mind. We defined a binary variable, $\zeta_{m,n}(t)\in \{0,1\}$, to indicate whether the UAV, m, collected dates from sensor n at time t.
(4)$$\zeta_{m,n}(t)=\left\{\begin{array}{l} 1\;m\;\mathrm{collect}\;\mathrm{information}\;\mathrm{from}\;n\\ 0\;\mathrm{otherwise} \end{array}\right.,$$

The maximum number of sensors that can be received by the UAV at any time is represented by the following constraint:(5)$$\sum_{n=1}^{N}\zeta_{m,n}(t)\leq K,$$

As the distance between the sensor and its associated UAV may be large, some sensors cannot upload all the data to the UAV within the limited energy range; this is due to the poor channel environment, so the UAV needs to transmit data within the limited range. During the flight of the UAV, the angle between the antenna and sensor changes continuously, so the direction of antenna needs to be adjusted continuously. Therefore, the UAV is divided into flight-hover modes in order to complete the data collection task.

Due to the space and mechanical limitations, the UAV has a maximum flight altitude, H, and a maximum velocity $v_{\max}$. Ignoring the purpose of exposition, the energy consumption caused by the acceleration/deceleration of the UAV, was similar to [22,26]. The UAV lifts off from the data center, then hovers at the collection position for data collection, and flies to the next collection position or back to the data center after data collection.

We assumed that the geographical location distribution characteristics of sensors are dense and sparse. When the sensors are relatively dense, the collection position of the UAV lies between the dense points, and the data of no more than K sensors can be collected at the same time. When the sensors are relatively sparse, the UAV may be sent to collect sensors in sparse locations. The set of sensors assigned to the UAV, m, is expressed as $N_{m}=\{N_{c_{1}},\ldots ,N_{c_{k}}\}$. The number of subsets, $N_{c_{k}}$, is $N_{m,}{}_{c_{k}}$, then $N_{m}=\sum_{k}N_{c_{k}}$. When $N_{c_{k}}>1$, it indicates that the sensors in set $N_{c_{k}}$ are dense. Otherwise, when $N_{c_{k}}=1$, it indicates that the sensors in set $N_{c_{k}}$ are sparse. The time of collection, $N_{c_{k}}$, regarding the UAV, m, is denoted as follows:(6)$$t_{m,c_{k}}=t_{m}^{N_{c_{k}},1}-t_{m}^{N_{c_{k}},0}+\delta ,$$
where δ is the time consumed for directional antenna adjustment. The collection time of the UAV, m, is expressed as follows:(7)$$t_{c_{m}}=\sum_{k}t_{m,c_{k}},$$

The data collection point of UAV, m, is denoted as $l_{m,}{}_{CP_{k}}$, and the flight time of the UAV, m, is denoted as follows:(8)$$t_{f_{m}}=\frac{\sum_{i=2}^{k}\left(\left\|l_{CP_{i}}-l_{CP_{i-1}}\right\|\right)+\left\|l_{BS}-l_{CP_{1}}\right\|+\left\|l_{BS}-l_{CP_{k}}\right\|}{\left\|v_{\max}\right\|}$$

### 3.5. Problem Formulation

The objective of this paper is to jointly optimize the task allocation and UAV flight path during UAV-assisted data collection, and to minimize the maximum time spent by UAVs within a collection cycle, under the constraints of meeting the fairness and priority criteria of data collection. The importance of the sensors is denoted by $G=\{\kappa_{1},\ldots ,\kappa_{n}\}$, among which, $\kappa_{i}\in \{0.5,1,1.5,2,2.5,3\}$. The amount of data to be collected from $n_{m}^{i}$ in a data collection cycle is denoted by $I_{n_{m}^{i}}$, and the fairness and priority constraints of data collection are denoted by the following:(9)$$\int_{t_{m}^{i,0}}^{t_{m}^{i,1}}R_{{\tilde{n}}_{m}^{i}}(t)\geq I_{{\tilde{n}}_{m}^{i}},$$
(10)$$\kappa_{n_{m_{1}}^{i_{1}}}:\kappa_{{\tilde{n}}_{m_{2}}^{i_{2}}}=I_{n_{m_{1}}^{i_{1}}}:I_{n_{m_{2}}^{i_{2}}},\;\;m_{1},m_{2}\in M,\;n_{1},n_{2}\in N,$$

According to Equation (10), the sensor with higher importance needs to upload more data. In this regard, the problems proposed in this paper are shown in P1, as follows:(11)$$P1\;\underset{\{\lambda_{n,m}\},\{\zeta_{m,n}(t)\},\{L_{m}\},m\in M_{d}\cup M_{s}}{\min}\max\;T_{m},$$
(12)$$s.t.\sum_{m=1}^{M}\lambda_{n,m}=1,n\in N,$$
(13)$$\lambda_{n,m}\in \{0,1\}$$
(14)$$\sum_{n=1}^{N}\zeta_{m,n}(t)\leq K$$
(15)$$\zeta_{m,n}(t)\in \{0,1\}$$
(16)$$\int_{t_{m}^{i,0}}^{t_{m}^{i,1}}R_{{\tilde{n}}_{m}^{i}}(t)\geq \;I_{{\tilde{n}}_{m}^{i}},m\in M,n_{m}^{i}\in N$$
(17)$$\int_{0}^{t_{f_{m}}+t_{c_{m}}}P_{f}(v)dt+t_{c_{m}}P_{c}\leq E_{\mathrm{th}},m\in M$$
(18)$$\kappa_{n_{m_{1}}^{i_{1}}}:\kappa_{{\tilde{n}}_{m_{2}}^{i_{2}}}=I_{n_{m_{1}}^{i_{1}}}:I_{n_{m_{2}}^{i_{2}}},\;\;m_{1},m_{2}\in M,\;n_{1},n_{2}\in N$$
where, $\lambda_{n,m}$ is an indicator variable used to represent the task assignment result. If the data of n is collected by m (i.e., $n\in N_{m}$), then $\lambda_{n,m}=1$, otherwise, $\lambda_{n,m}=0$. That $\sum_{m=1}^{M}\lambda_{n,m}=1$ means a ground sensor can only be accessed using a UAV. The constraint (12)–(15) is an integer constraint, representing the correlation between the UAV and the sensor; (16) and (17) is an integral constraint, representing the constraint on the amount of data collected by the UAV and the constraint on the energy consumption of the UAV, respectively; (18) is an equality constraint, representing the relationship between the importance of the sensor and the amount of upstream data.

In order to obtain the optimal solution of problem P1, N sensors must be assigned to M UAVs, so that each UAV can perform as few tasks as possible within the energy budget. The problem is extremely challenging for three reasons. First of all, this problem needs to optimize three variables, among which, $\lambda_{n,m}$ and $\zeta_{m,n}(t)$ are discrete variables, and $L_{m}$ represents the continuous variables. They have coupling relations with each other, so it is difficult to optimize at the same time. Secondly, the optimization variables, $\lambda_{n,m}$ and $\zeta_{m,n}(t)$, concerning the communication scheduling between the UAV and sensor, are binary, so the purpose problem is a non-convex problem. Thirdly, the task time is composed of two parts, as follows: flight task and collection task. The minimum sum of the flight task and collection task should be taken into consideration during optimization, and the hovering collection position of the UAV affects the time consumption of both the task and collection task. It is very difficult to find the best collection position and the task assignment scheme of the UAV, traversing all collection points and collection sequences. In the existing literature, the UAV task allocation scheme based on sensor density has not been considered. Therefore, this paper proposes AASDC to obtain the suboptimal solution of UAV task allocation, and it further discusses the problem of data collection in the case of sensor density deployment.

## 4. AASDC

The proposed AASDC comprises two steps. Firstly, we converted the task assignment of UAVs into a graph cut problem, and we used the normalized cut(Ncut) algorithm to assign the tasks of the UAV. Secondly, for the data collection of a single UAV, including hovering position selection and trajectory planning, the Density-Based Spatial Clustering of Applications with Noise (DBSCAN) and particle swarm optimization (PSO) algorithms were combined to obtain the hovering position of the UAV. Then, the flight path problem of the UAV’s hovering position was transformed into a TSP problem, which was solved using the ant colony algorithm (ACA).

### 4.1. UAV Task Assignment

The task assignment problem of multiple UAVs is a multi-objective combinatorial optimization problem, therefore, it is not easy to obtain the optimal solution. We naturalized the task assignment problem into a graph cut problem to establish the corresponding objective function with a partition on the graph, and the minimum value of the objective functions was found to correspond with an optimal grouping on the image. More specifically, the sensors were regarded to be the vertices on the graph, and the similarity between the sensors was regarded to be the weights of the edges between vertices, so an undirected weighted graph based on the sensors’ similarities is obtained as G=(V,E,S). The node set of G is $V=\{v_{1},v_{2},\ldots ,v_{n}\}$, the set of edges connecting nodes is E, and the Gaussian similarity function, $s_{ij}=\exp(-d^{2}(v_{i},v_{j})/2\sigma_{1}{}^{2})$, was used to measure the similarity between $v_{i}$ and $v_{j}$. The symmetric similarity matrix is expressed as $S=(s_{ij})$. If the set of nodes, V, is divided into two independent subsets, A and B, where B=V−A, removing the edge connecting all nodes in A and B gives a degree of separation between A and B, denoted as cut(A,B).
(19)$$cut(A,B)=\sum_{i\in A,j\in B}s_{ij},$$

Wu and Leahy proposed Mincut based on the minimum partition criterion [30], but Mincut may easily partition isolated points in the graph. To overcome this phenomenon, Shi and Malik proposed that Ncut describes the degree of separation between two clusters and the degree of separation between multiple clusters [31], defined as follows:(20)$$Ncut(A,B)=\frac{cut(A,B)}{assoc(A,V)}+\frac{cut(B,A)}{assoc(B,V)},$$
(21)$$Ncut_{k}=\frac{cut(A_{1},V-A_{1})}{assoc(A_{1},V)}+\frac{cut(A_{2},V-A_{2})}{assoc(A_{2},V)}+\cdots +\frac{cut(A_{k},V-A_{k})}{assoc(A_{k},V)},$$
where, $assoc(A,V)=\sum_{i\in A,j\in V}s_{ij}$ is the sum of the connection weights of nodes in *A* and all nodes in the graph; $A_{i}$ is the i th subset of V. Ncut overcomes the minimum cut criterion to easily divide the isolated points in the graph, and it exhibits a good clustering performance. The partition corresponding to the smallest Ncut value is the optimal partition of the graph, G. Under the constraint of the relaxation optimization objective function, the discrete optimization problem of minimizing the Ncut value is transformed into the following:(22)$$D^{-1/2}(D-S)D^{-1/2}\xi =\lambda \xi ,$$
where D is the diagonal matrix of n×n, whose diagonal elements are $d_{ii}=\sum_{j}s_{ij}$, and λ and ξ are the corresponding eigenvalues and eigenvectors, respectively. When the cluster number is two, the feature vector corresponding to the second smallest feature value of Equation (22) is used to optimally partition the graph, G, and the segmentation result of the corresponding image is obtained. When the cluster number is k(k>2), the feature vector corresponding to k, and the largest eigenvalues of Equation (22), are used for a series of processing to obtain clusterification. As shown in Algorithm 1, we use the Ncut algorithm to cluster the sensors.
**Algorithm 1:** Ncut algorithm.**Input:**
The vertices set V={v1,v2,…,vn}, vi∈RD,
      k,σ1: parameter, k=M**Output:** A set of k clusters Nm={Nc1,…,Nck}
 1: Defined distance d(vi,vj)=vi−vj2, calculate symmetric similarity matrix S=(sij), sij=exp(−d2(vi,vj)/2σ12), if i≠j, and sii=0;
 2: Calculate matrix D and matrix L=D−1/2(S)D−1/2;
 3: Calculate the eigenvector u1,u2,…,uk corresponding to k maximum     eigenvalue of matrix L, denote U0=[u1,u2,…,uk]∈ℝn×k;
 4: Each row of matrix U0 is unitized to be matrix U
(i.e.,Uij=U0/(∑jU02ij)1/2);
 5: Each row of matrix U is regarded as a point in ℝk, and these points are divided into cluster k by K-means clustering algorithm
 6: Finally, assign the original point vi to cluster j if and only if row i of the matrix U was assigned to cluster j. 

During the process of sensor clustering, Algorithm 1 can minimize the value of $Ncut_{k}$, that is to say, the global similarity of the k clusters is minimized. According to the similarity fraction, when $s_{ij}=\exp(-d^{2}(v_{i},v_{j})/2\sigma_{1}{}^{2})$ is smaller, the distance between sensors of different clusters is larger, and the distance between sensors of the same cluster is smaller. Therefore, this algorithm fully considers the distance relationship between sensors, so that the distance between the sensors of each cluster is smaller. By assigning the data collection tasks of each cluster sensor to a single UAV, the flight distance of the UAV can be reduced, and thus, the collection time can be shortened.

The complexity of the Ncut algorithm mainly depends on the size of the graph and the number of iterations. The Ncut algorithm is a graph partitioning algorithm that is used to divide nodes in a graph into disjoint subsets. The approximate complexity of the Ncut algorithm is as follows: 1. Construct a similarity matrix, O($n^{2}$); 2. Solve for the eigenvalues and eigenvectors, O($n^{3}$) (this step requires the eigenvalue decomposition of the similarity matrix, and the computational complexity is high); and 3. Iterative optimization partitioning results. It is usually necessary to carry out multiple iterations, and the computational complexity of each iteration is O($n^{2}$). The number of iterations is usually small, but it depends on the size of the problem and the convergence conditions. Overall, the time complexity of the Ncut algorithm is O($n^{3}$), where n is the number of sensors.

### 4.2. UAV Hover Position Selection and Trajectory Optimization

In Algorithm 1, sensors are divided into groups, and each group of sensors is assigned to a single UAV to complete tasks independently. In this section, we propose a hybrid algorithm for DBSCAN, PSO, and hovering position data collection to optimize the hovering position of the UAV; then, we adopted the ant colony algorithm to obtain the optimal flight trajectory of the UAV. The DBSCAN algorithm is used to classify sensors, as described in Algorithm 2.
**Algorithm 2:** A density-based clustering algorithm.**Input:**
  *D*: a data set containing *n* objects,
  *ε*: the radius parameter, and
  *MinPts*: the neighborhood density threshold.
**Output:** A set of density-based clusters
 1: mark all objects as unvisited;
 2: **do**
 3:   randomly select an unvisited object *p*;
 4:   mark *p* as visited;
 5:   **if** the *ε*-neighborhood of *p* has at least *MinPts* objects
 6:    create a new cluster *C*, and add *p* to *C*;
 7:    let *N* be the set of objects in the *ε*-neighborhood of *p*;
 8:    **for** each point *p*′ in *N*
 9:      **if**
*p*′ is unvisited 
 10:      mark *p*′ as visited;
 11:      if the *ε*-neighborhood of *p*′ has at least *MinPts* points, add those points to *N*;
 12:      if *p*′ is not yet a member of any cluster, add *p*′ to *C*;
 13:     **end for**
 14:     output *C*;
 15:    **else** mark *p* as noise;
 16:   **end if**
 17: **until** no object is unvisited;
 18: **end**
 19: **end**

Algorithm 2 clusters the sensor set into core points and noise points. The core points are clustered together and the noise points are sparsely distributed in the area. Thus, we regarded the classified core points as dense sensors, and the noise points as sparse sensors. Then, for set $N_{m}=\{N_{c_{1}},\ldots ,N_{c_{k}}\}$, the clustering of core points is classified into a sub-set, and each noise point is separately classified into a sub-set. Thus, the UAV, m, has k hovering points in a mission.

The complexity of the DBSCAN algorithm is O($\max(N_{m})\log(\max(N_{m}))$), where $N_{m}$ is the size of the data set.

Different data collection methods are adopted in this section, in accordance with the core points and noise points. For $N_{c_{k}}>1$, according to Equations (3), (6), and (7), the target problem can be expressed as per P2.
(23)$$P2\;\underset{\left\{l_{CP_{k}}\right\}}{\min}\{t_{m,c_{k}}\},$$
(24)$$s.t.\zeta_{m,n}(t)\in \{0,1\},$$
(25)$$\sum_{n=1}^{N}\zeta_{m,n}(t)\leq K$$
(26)$$\int_{t_{m}^{i,0}}^{t_{m}^{i,1}}R_{{\tilde{n}}_{m}^{i}}(t)\geq \;I_{{\tilde{n}}_{m}^{i}},\forall m\in [1,\ldots ,M-1],n_{m}^{i}\in \{N_{1},N_{2},\ldots ,N_{M-1}\}$$
(27)$$\int_{0}^{t_{f_{m}}+t_{c_{m}}}P_{f}(v)dt+t_{c_{m}}P_{c}\leq E_{\mathrm{th}},m\in M,$$
(28)$$\kappa_{n_{m}^{i_{1}}}:\kappa_{{\tilde{n}}_{m}^{i_{2}}}=I_{n_{m}^{i_{1}}}:I_{n_{m}^{i_{2}}},\;\;m\in M,\;n_{1},n_{2}\in N,$$

Problem P2 regards the hovering position of UAV as an independent object, and it optimizes the position of the UAV to minimize the task time of the UAV. Since the hovering position of the UAV can be any point in the plane space, it is difficult to find the optimal solution, so the particle swarm optimization algorithm is adopted to find the sub-optimal solution. The description is shown in Algorithms 3 and 4.
**Algorithm 3:** The location of the UAV based on PSO.**Input:**
Each UAV collects the sensors set, $N_{c}$, the corresponding amount of data, $I_{n_{m}}(n_{m}\in N_{c})$, and the collection position $l_{m}$(∀m∈M).
**Output:** $l_{m}$

 1: **for**
m=1 to M−1
 2:   **for** each particle i∈Num;
 3:   Initialize velocity $V_{i}$ and position $X_{i}$ for particle i;
 4:   Evaluate particle i and set $pBest(i)=X_{i}$;
 5:   **end for**
 6:  gBest=min{pBest(i)};
 7:   **while** not stop
 8:    **for** i∈Num
 9:     Update the velocity and position of particle i;
 10:    Evaluate particle i;
 11:     **if** $fit(X_{i})<fit(pBest(i))$
 12:     $pBest(i)=X_{i}$;
 13:     **if**
fit(pBest)<fit(gBest)
 14:     gBest=pBest(i);
 15:     **end for**
 16:   **end while**
 17:  $l_{m}=gBest;$
 18: **end**
 19: **end**

**Algorithm 4:** Greedy algorithm for UAV data collection.**Input:**
The value of $N_{m},m\in [1,\ldots ,M-1]$, the number of channel K, the coordinates set of $l_{n_{m}}(n_{m}\in N_{m})$, the coordinate of base $l_{BS}$, UAV collection position $l_{m}$
**Output:** $t_{c_{m}},m\in [1,\ldots ,M-1]$

 1: Calculate the time taken $t_{n_{m}}$ by the UAV m to collect sensor $n_{m}$, 
 2: Order $\left\{t_{n_{m}},n_{m}\in N_{m}\right\}$ from largest to smallest, and the sequence is $T_{m}$,
 3: **if** K=1 
 4:  $t_{c_{m}}=\sum_{n_{m}\in N_{m}}t_{n_{m}}$,
 5: **else if**   K>N
 6:  $t_{c_{m}}=\max\{t_{n_{m}}\}$,
 7: **else if**   K<N
 8:   **for** i=1:K
 9:    $t_{i}=T_{m}^{i}$,
 10:   **end for**
 11:   **for** i=K+1:N
 12:    $t_{c_{m}}=\max\{t_{i},\min\{t_{i}\}+T_{m}^{i}\}$,
 13:   **end for**
 14: **end**
 15: **end**

In Algorithm 2, the updated formula of particle velocity is expressed as follows:(29)$$V_{i}=w*V_{i}+c_{1}r_{1}(pBest(i)-X_{i})+c_{2}r_{2}(gBest-X_{i}),$$

The update formula for location is expressed as follows:(30)$$X_{i}=X_{i}+V_{i},$$

The fitness is as follows:(31)$$fit(i)=t_{m,c_{k}}(i),$$
where $t_{m,c_{k}}$ is based on Equation (6), and $t_{c_{m}}$ is an UAV in the coordinate of $l_{CP_{k}}$ to collect the task time of $N_{c_{k}}$ sensors, as shown in Algorithm 4.

The complexity of the PSO algorithm is usually O(iter1*Num), where iter1 is the number of iterations and Num is the number of particles.

The computational complexity of the greedy algorithm for UAV data collection is O($(\max N_{m})\log N_{m}$), because in the greedy algorithm, we need to sort the fetching time in order to select the sensors in the queue, according to the optimal strategy. The time complexity of sorting is O($(\max N_{m})\log N_{m}$), where $n_{m}$ is the number of sensors. Then, we need to iterate the sorted queue, calculating each person’s wait time in turn. The time complexity of traversal is O($\max N_{m}$). Therefore, the computational complexity of the entire greedy algorithm is O($(\max N_{m})\log N_{m}$), where $N_{m}\ll \max(N_{m})$.

When the distribution of sensors in $N_{M}$ is sparse, and the UAV cannot collect multiple sensors at a suspension point, and then send the Mth UAV to collect sensors’ data in the fly-hover mode. The collection mission is based on (7) and (8), which is shown as P3.
(32)$$P3\;\underset{\left\{L_{m}\right\}}{\min}\{t_{c_{m}}+t_{f_{m}}\},$$
(33)$$s.t.\zeta_{M,n}(t)\in \{0,1\},$$
(34)$$\sum_{n=1}^{N}\zeta_{M,n}(t)\leq 1,$$
(35)$$\int_{t_{m}^{i,0}}^{t_{m}^{i,1}}R_{{\tilde{n}}_{m}^{i}}(t)\geq \;I_{{\tilde{n}}_{m}^{i}},n_{m}^{i}\in N_{M},$$
(36)$$\int_{0}^{t_{f_{M}}+t_{c_{M}}}P_{f}(v)dt+t_{c_{M}}P_{c}\leq E_{\mathrm{th}},$$
(37)$$\kappa_{n_{m}^{i_{1}}}:\kappa_{{\tilde{n}}_{m}^{i_{2}}}=I_{n_{m}^{i_{1}}}:I_{n_{m}^{i_{2}}},\;\;m\in M,\;n_{1},n_{2}\in N,$$

In problem 3, there are a total of k hovering points of $L_{m}$. Since the value of k is related to the parameters, MinPts, and ε in Algorithm 2, different parameter combinations have a great impact on the final clustering effect. Therefore, we use the ergodic method combined with the ACA to obtain the optimal result within the range of the distance threshold and sample threshold. The problem, P3, can be factored into the TSP problem, using the ant colony algorithm, as shown in Algorithm 5.
**Algorithm 5:** UAV data collection trajectory based on the ant colony algorithm.**Input:** The value of $N_{M}$, the coordinates set of $l_{n_{m}}(n_{m}\in N_{m})$, the coordinate of base $l_{BS}$
**Output:** $t_{c_{m}}+t_{f_{m}}$, UAV data collection sequence 
 1: **for**
MinPts in range (20,120)

 2:   **for** ε in range (2,6)

 3:    update **Algorithm 2**
 4:    **for** each edge, 
 5:     set initial pheromone,
 6:    **end for**
 7:    **while** not stop
 8:     **for** each ant
 9:      choose an initial city $l_{BS}$,
 10:      **for** i=1 to k
 11:      choose next city j with the probability given in Equation (38),
 12:      **end for**
 13:     **end for**

 14:     compute the $t_{m}$ of the tour constructed by the kth ant by 
      **Algorithm 3**, **Algorithm 4** and Equations (3), (6)–(8) 
 15:     **for** each edge
 16:      update the pheromone value in Equation (39),
 17:     **end for**
 18:    **end while**
 19:    get $t_{m}$ in Equation (8)
 20:   update $\min\;t_{m}$

 21:   **end for**
 22: **end for**

The probability of ant k from i to j is as follows:(38)$$P_{ij}^{k}=\left\{\begin{array}{l} \frac{\tau_{ij}^{\alpha }(t)*\eta_{ij}^{\beta }(t)}{\sum_{s\in allowed_{k}}\tau_{ij}^{\alpha }(t)*\eta_{ij}^{\beta }(t)}\;\;\;\;j\in allowed_{k}\\ 0\;\;\;\;\;\;\;\;\;\;\;\;\;\;\;\;\;\;\;\;\;\;\;\;\;\;\;\;\;\;\;\;otherwise \end{array}\right.,$$

In this case, i and j are the starting point and the end point, and $\eta_{ij}(t)=1/d_{ij}$ is used to represent the visibility of the ant from i to j, which is the size of the path distance from i to j. The $\tau_{ij}(t)$ is the information concentration of the i to j, and the $allowed_{k}$ represents the set of nodes that ant k has not yet visited.

The formula for updating pheromones is as follows:(39)$$\tau_{ij}(t+1)=\tau_{ij}(t)*(1-\rho )+\Delta \tau_{ij},0<\rho <1,$$
(40)$$\Delta \tau_{ij}=\sum_{k=1}^{m}\Delta \tau_{ij}^{k}$$
(41)$$\Delta \tau_{ij}^{k}=\left\{\begin{array}{l} {\left(C_{k}\right)}^{-1},(i,j)\in R^{k}\\ 0,\;\;\;\;\;\;\;otherwise \end{array}\right.,$$

The computational complexity of ACA mainly depends on the problem size and the number of iterations. In the TSP problem, assuming there are $N_{m}$ cities, the time complexity of the ant colony algorithm is roughly O(${\left(\max N_{m}\right)}^{2}*iter2$), where iter2 is the number of iterations. In sum, the complexity of the AASDC is upper-bounded by O($n^{3}{\left(\max N_{m}\right)}^{3}\log(\max N_{m})\cdot iter2\cdot iter1\cdot Num$).

## 5. Simulation Results

In this section, we present the progressive analysis of the AASDC. Without a special statement, we considered four UAVs that serve 50 sensors. The UAVs have a constant flying altitude, H, and the sensors’ transmit power was fixed at P. The proposed scheme was simulated using matlab2021 on a platform with CPU i7-8750h and a memory of 16 GB. The other simulation parameters are listed in Table 3.

To illustrate the performance of the proposed scheme, we compared our proposal with three schemes, as follows:(1)MTSP-based algorithm (MTSP): This algorithm designs the task assignment of multiple UAVs by using the min–max Multiple Traveling Salesman problem (MTSP) algorithm.(2)Immune genetic and PSO algorithm (IGPA): This algorithm designs the task assignment of multiple UAVs using the choice address problem, and the algorithm solves it by using the immune genetic algorithm, then it optimizes the hovering point of the UAV using the particle swarm optimization (PSO) algorithm.(3)Kmeans-based algorithm (KMEANS): This algorithm uses the kmeans algorithm to obtain the task assignment of multiple UAVs. Then, it uses ACA to obtain the flight trajectory of each UAV.

### 5.1. Different Sensors’ Distribution Characteristics

As shown in Figure 2, we designed three different distributed characteristics of sensors to compare the effect of the algorithm. The number and data size of the sensors are fixed. The distribution of sensors in Figure 2a–d is characterized by one center and one sub-center, in which, 30 sensors are deployed in the central area and 20 sensors are deployed in the sub-center area. The distribution of sensors in Figure 2e–h is characterized by four central modes with 12 or 13 sensors deployed in each area, whereas the distribution of sensors in Figure 2i–l involves random deployment in the collection area. Figure 2a–l reflects the data collection effects of AASDC, MTSP, IGPA, and KMEAN, respectively, using three distributions.

As shown in Figure 3, we compared the time required for the UAV to collect the same amount of data under different sensor distribution characteristics, where the amount of data for each sensor is 0.5 Mb. When the AASDC algorithm is adopted, the collection time of three different sensor distribution scenarios reaches their minimum value. When sensors were distributed in the center and sub-center, the collection time was reduced by more than 10%. Therefore, for some non-random distribution scenarios, the proposed scheme can shorten the UAV data collection time.

### 5.2. Different Data Sizes

As shown in Figure 4, using different collection data sizes, we compared the average collection time of the proposed scheme and the comparison scheme using the proposed three sensor distributions. The horizontal coordinate is the amount of data comprising all sensors, and as the amount of data collected increases, so does the average acquisition time of the task’s completion time. When the data collected are less than 3 Mb, the collection time difference between algorithms is not obvious. However, when the data volume is greater than 9 Mb, the collection time of the IGPA algorithm increases rapidly, and the collection time of the MTSP and K-MEANS algorithms is slightly higher than that of the AASDC algorithm. When the amount of data collected reaches 30 Mb, the collection time of the MTSP algorithm and K-MEANS algorithm is about 10% higher than that of the AASDC algorithm. In the comparison algorithm, the KMEANS algorithm only performs clustering depending on the sensor location, without considering the influence of the sensor’s data and collection time. Although the IGPA algorithm considers the position of the sensor and the amount of data comprising the sensor, it takes a long time to collect the data of a few distant sensors in the hovering position of the UAV. The MTSP algorithm emerges directly above the sensor to collect data, ignoring that when the sensors are concentrated, the UAV can choose to collect data at a hovering point to reduce the collection time and flight energy consumption. The proposed AASDC distinguishes between dense points and sparse points during the collection process. The dense points do not move the position of the UAV, and the sensor in the sparse points emerges directly above to collect data; they can also distinguish between the distribution characteristics of the sensors to a greater extent. Therefore, the collection time of the AASDC is smaller than that of other schemes.

As shown in Figure 5, we compared the maximum energy consumption and average energy consumption of the proposed scheme with the comparison scheme for UAV use using different data collection systems. As can be seen from Figure 5a,b, when the proposed scheme collects the same data, the energy consumption of the UAV is less than that of other schemes. The difference in energy consumption between AASDC, IGPA, and KMEANS is not obvious, because there is little difference between the flight energy consumption and the collection energy consumption of the UAV at this time. However, the collection energy consumption of the UAV gradually increases with the increase in the amount of data collected. Compared with other schemes, AASDC adopts the method of simultaneously collecting data from multiple sensors in the area where sensors are concentrated. For the area wherein sensor distribution evacuation occurs, the sensor is flown directly above the point where data collection takes place, and condensation takes into account the distribution characteristics of the sensor, so the UAV energy consumption is reduced.

## 6. Discussion

### 6.1. The Tradeoff between the Distance between the UAV and the Sensor and the Flight Distance

In this section, we discuss the impact of the distance between the UAV and the sensor mission time of the UAV during UAV-assisted data collection. In the wireless network of the UAVs, in order to maximize the communication rate, the UAV always flies close to the sensor to improve the capacity of the link and to reduce the collection time of the UAV. It has also led to an increase in flight time. Therefore, there is a tradeoff between the distance flown by the UAV and the distance between the UAV and the sensors. In this paper, using the classification of sensor density characteristics, we focus on the problem of minimizing the sum of the flight time and collection time when multi-antenna and multi-channel UAV auxiliary communication meets the requirements of data collection. In the future, we will consider these tradeoffs further.

### 6.2. Multi-UAV Task Allocation

Unlike ground-based communication vehicles, which are restricted in terms of their trajectories and hovering positions during auxiliary communication, unmanned aerial vehicles (UAVs) have the ability to hover and move freely in any position in the air under ideal conditions. Consequently, cooperative-assisted communication between multiple UAVs is more intricate. In this paper, we studied the task assignment of multiple UAVs with fixed heights and specific sensor distribution characteristics. However, in future research, using more complex scenarios involving sensor distributions, it will be necessary to consider how to construct data models and obtain more accurate task assignment schemes.

### 6.3. The Theoretical and Practical Implications

For UAV-assisted data collection, in this paper, the characteristics of sensor distribution, such as the geographical position and importance of the sensor, are discussed. We provided an effective data collection scheme based on sensor distribution characteristics. Regarding practical applications for sensor distribution, they are relatively concentrated and partially dispersed in scenarios such as disaster areas, where the proposed scheme exhibits a good effect. When the number of sensors is large and the distribution is very sparse, the application of the algorithm will be limited, which needs to be improved in future work.

## 7. Conclusions

In this paper, we considered multi-UAV-assisted data collection WSNs, in which the UAV is equipped with multiple directional antennae. Considering the geographical location distribution characteristics and importance of sensors, the goal of minimizing the data collection time was achieved by optimizing the trajectory, hovering position, and task allocation of the UAV. In this regard, the AASDC scheme proposed in this paper can save time and energy using the same data collection size.

Since we added some constraints to the model, there are certain limitations that occur in practice. For example, we assumed that UAVs can only collect data in hover mode. In fact, when considering the data collected during the flight of UAVs, the collection trajectory changed. In the future, we will build multi-antennae UAVs in a flight mode data collection model to further improve the performance of multi-UAV data collection systems. In addition, this paper also discusses the importance of sensors in WSNs, and the influence of geographic information featured at the location, and collected by UAS. However, the features of the sensors can be divided into more types. For example, the logic-based framework proposed in [32] can be extended and applied to the construction of WSNs. Building the characteristics of sensor networks in a manner that is more in line with the application of specific scenarios will be studied and discussed in the next step. 

## Figures and Tables

**Figure 1 sensors-23-07842-f001:**
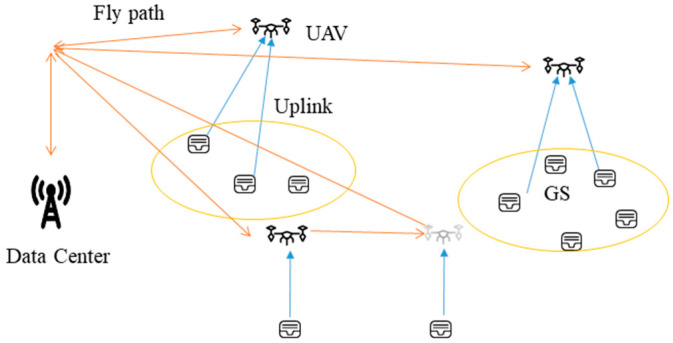
System model.

**Figure 2 sensors-23-07842-f002:**
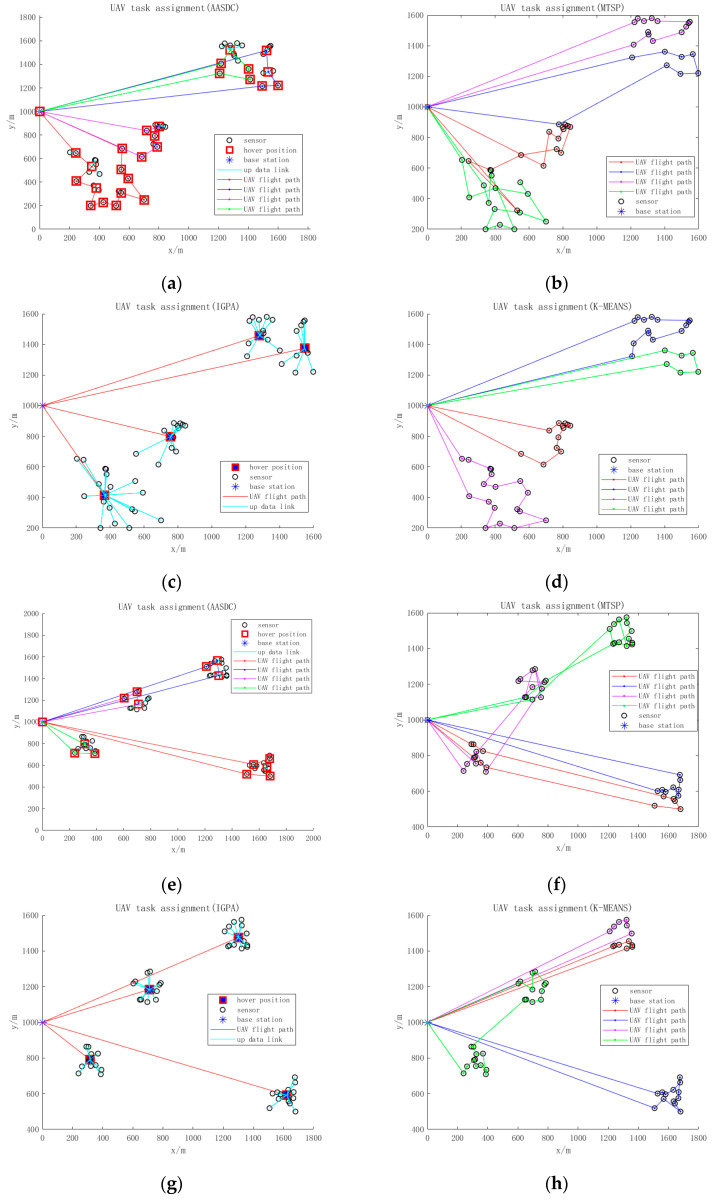
(**a**–**l**) reflects the data collection effects of AASDC, MTSP, IGPA, and KMEANS, respectively, using three distributions.

**Figure 3 sensors-23-07842-f003:**
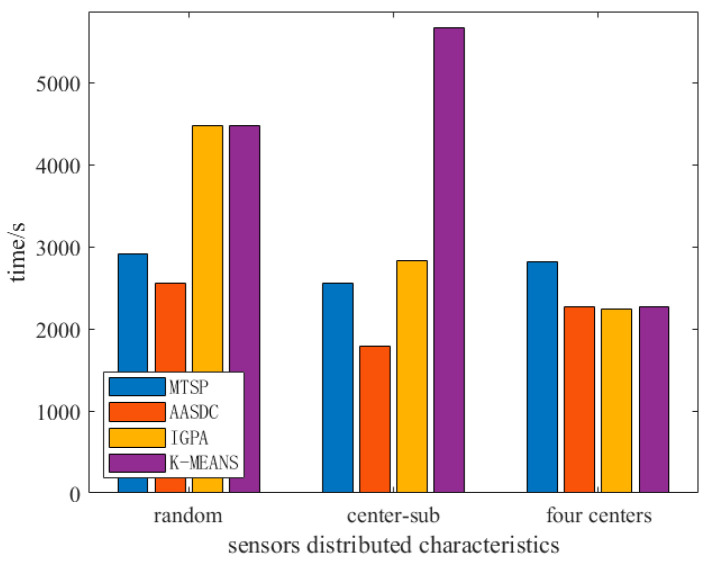
Different sensors’ distributed characteristics.

**Figure 4 sensors-23-07842-f004:**
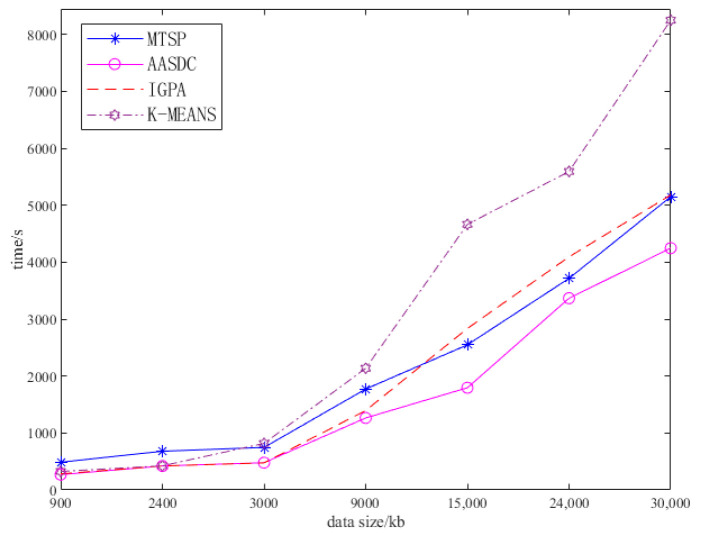
Different data sizes.

**Figure 5 sensors-23-07842-f005:**
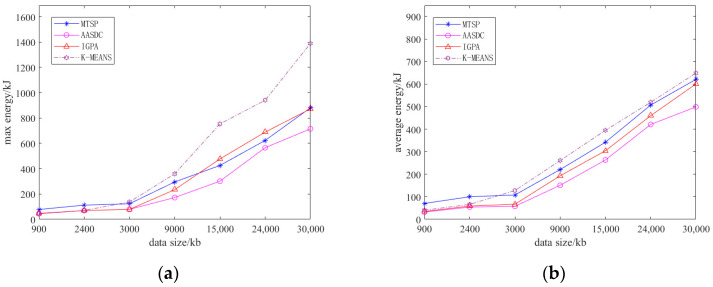
Maximum energy (**a**) and average energy (**b**) consumption in different data sizes.

**Table 1 sensors-23-07842-t001:** Comparison of related works with our work.

Reference	Objective Function	Decision Variables	Constraints
[18]	Min–max of flight path of UAVs	UAV path planning, UAV task assignment	Communication radius
[19]	Minimization of flight time of UAVs	UAV task area allocation, UAV path planning	Speed constraint, guaranteeing a successful data collection
[21]	Minimization of multi-UAV flight time of data collection	UAV path planning, UAV-sensor association scheme	Data upload constraint, energy constraint
[22]	Minimization of multi-UAV flight time regarding data collection	UAV path planning	Energy constraint of UAV, uplink throughput
[23]	Minimizing the deployment cost of UAVs	UAV path planning	Delay to sensitive data collection
[24]	Minimization of multi-UAV flight time regarding data collection	UAVs trajectories, and hovering and flying communication scheduling	Communication radius, velocity of UAV
[25]	Min–max of mission completion time among all UAVs	UAVs trajectories, wake-up scheduling and association for SNs	Upload a target amount of data with a given energy budget
[15]	Minimization of the task completion time of all the UAVs	UAVs trajectories	Energy budget
[26]	Minimization of the number and total operation times of the UAVs	the UAV trajectory and hovering location.	UAV’s energy budget and cache capacity, the data transmission constraints
[27]	Maximization of the throughput of the UAVs	hovering location, UAV-sensor association scheme	Different importance levels and quality of service requirements of sensors
[13]	Maximum and minimum data collection rate	UAV time allocation, UAV 3-D path planning	Energy consumption
[28]	Minimization of flight length of UAV	UAV collision avoidance, UAV path planning	Avoiding collisions
our work	Minimization of the task completion time of all the UAVs	UAV path planning, UAV hovering point, UAV task assignment	Energy budget of the UAV, the importance of sensors, the sensors’ distribution characteristics

**Table 2 sensors-23-07842-t002:** Major notations.

Notation	Definition
M	UAV set
N	Sensor set
m	UAV m
n	Sensor n
M	Number of UAVs
N	Number of sensors
K	Number of antennas, or number of channels
$l_{m}(t)$	Trajectory of m
$l_{n}$	Position of n
$l_{BS}$	Position of Base Station
$d_{{\tilde{n}}_{m}^{i},m}$	Distance between UAV m and sensor ${\tilde{n}}_{m}^{i}$
$V_{\max}$	Maximum velocity
$h_{{\tilde{n}}_{m}^{i},n}$	Instantaneous channel power gain
$R_{{\tilde{n}}_{m}^{i}}(t)$	Instantaneous channel capacity
$E_{\mathrm{th}}$	Energy threshold of UAV
$P_{f}(v)$	Flight power of the UAV at speed v
$P_{c}$	Circuit power consumed by UAV during data collection
$I_{n}$	Total amount of data of n
B	Bandwidth
$P_{{\tilde{n}}_{m}^{i}}$	Transmission power
$\sigma_{0}{}^{2}$	White Gaussian noise power
H	Altitude of UAV
$t_{f_{m}}$	Flying time of m
$t_{c_{m}}$	Hovering time m
$T_{m}$	The data collection period of m

**Table 3 sensors-23-07842-t003:** Simulation parameters.

Definition	Parameters	Values
Collection area size	D	2 km × 2 km
Altitude of UAV	H	50 m
Maximum velocity	$v_{\max}$	18 m/s
Bandwidth	B	1 MHz
Unit channel power gain	$\beta_{0}$	−50 dBm
White Gaussian noise power	$\sigma^{2}$	−110 dBm
Transmission power	P	5 dBm
Position of Base Station	$l_{BS}$	(1000, 0)
Energy threshold of UAV	$E_{\mathrm{th}}$	1000 kJ

## Data Availability

The data presented in this study can be requested from the authors.

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
