# Peer review of "Task Assignment Optimization in Multi-UAV-Assisted WSNs Considering Energy Budget and Sensor Distribution Characteristics"

_sensors, 2023, doi:10.3390/s23187842_

Round 1
Reviewer 1 Report
In this paper an optimization problem is proposed to minimize the time of each task in the periodic collection task of UAVs when the sensors have distribution characteristic and minimizes the task time of UAVs by optimizing the task assignment, trajectory and deployment position of UAVs. I have the following concerns.
1. Please rewrite the paper abstract to be more directive and to contain work results.
2. The limitations of the proposed optimization approach should clearly be discussed.
3. The figures in the paper need to be clearer and more legible. Please provide a clearer version of the obtained results.
4. Please divide Fig. 4 into three figures one for each distributed characteristic.
5. The paper conclusion is not written in a good way. Please enhance the conclusion by incorporating a quantitative summary of the results for a more comprehensive understanding.
6. Please update the paper refences, 2018 and above.
7. Future work must be included at the end of paper conclusion.
8. Please check the equations numbered (Eq. no. 9 not exist).
1. Moderate editing of the English language is necessary. There are many typos (Eq.9 not exist).
Reviewer 2 Report
Author investigated the collaborative task allocation for data collection in a multi-UAV-assisted sensor network. However, before acceptance, minor corrections are required:
1. In abstract, please clearly specify the Purpose, Contribution, and findings.
2. Please check the Algorithm 1.
3. Please check the Algorithm 2. End statement missing.
4. Please check the Algorithm 3. End statement missing.
5. Please check the Algorithm 5. Line 1 and 2 are not clear. 1: for MinPtsÎ[20,120] 2: for e Î[2,6]
6. Please analysis the complexity in your model.
7. Which software you used for simulation.
8. Please give some future works end of conclusion.
9. It is recommended to use a professional proofread and native English correction.
It is recommended to use a professional proofread and native English correction.
Reviewer 3 Report
This paper presents an interesting approach to task assignment optimization in Multi-UAV-assisted WSNs. The goal is to minimize task completion time by optimizing task assignment, trajectory planning, and deployment positions of UAVs, while considering sensor distribution characteristics and energy constraints. The proposed approach divides the problem into task assignment and trajectory optimization sub-problems, introducing an algorithm based on sensor distribution characteristics (AASDC) for solving it. Simulation results demonstrate that the proposed scheme outperforms existing methods in terms of data collection time for the same amount of data.
While the work presented is valuable and the experiments carried out are extensive, there are several major points that require additional clarification and detail for improved understanding, transparency, and reproducibility of the study.
- It is necessary to highlight the contribution in a more solid manner.
- Related work section should be expanded.
- Starting from the previous works, I suggest introducing a table to summarize the most recent works and to highlight the novelty of the proposed work.
- What are the limitations of the proposed study?
- The discussion of the results should better highlight the novelty of the proposed study and the evaluation performed.
- Although the method is described soundly and the experimental evaluation is solid, the implications are not very clear. I suggest the authors to provide a discussion about the theoretical and practical implications of such work.
Also, it would be beneficial for the authors to cite the following paper to provide a more comprehensive context for their work: doi.org/10.1017/S1471068419000449. This reference proposes a logic-based framework to simulate the evolution of complex systems, which can be extended to WSNn.
Round 2
Reviewer 1 Report
The paper can be accepted in its current form. But one more concern about paper figures. The figures in the paper need to be clearer and more legible. Please provide a clearer version of the obtained results especially Fig. 4.
Moderate editing of the English language is necessary.
Reviewer 2 Report
Thanks for correction.
Well done.
Reviewer 3 Report
The authors successfully addresse my concerns. There are no other comments from my side.